# Magic Leap 1 versus Microsoft HoloLens 2 for the Visualization of 3D Content Obtained from Radiological Images

**DOI:** 10.3390/s23063040

**Published:** 2023-03-11

**Authors:** Giulia Zari, Sara Condino, Fabrizio Cutolo, Vincenzo Ferrari

**Affiliations:** 1Information Engineering Department, University of Pisa, Via Girolamo Caruso, 16, 56122 Pisa, Italy; g.zari@studenti.unipi.it (G.Z.); sara.condino@unipi.it (S.C.); vincenzo.ferrari@unipi.it (V.F.); 2EndoCAS Center, Department of Translational Research and New Technologies in Medicine and Surgery, University of Pisa, 56126 Pisa, Italy

**Keywords:** mixed reality, medical imaging, Microsoft HoloLens, Magic Leap, head-mounted displays

## Abstract

The adoption of extended reality solutions is growing rapidly in the healthcare world. Augmented reality (AR) and virtual reality (VR) interfaces can bring advantages in various medical-health sectors; it is thus not surprising that the medical MR market is among the fastest-growing ones. The present study reports on a comparison between two of the most popular MR head-mounted displays, Magic Leap 1 and Microsoft HoloLens 2, for the visualization of 3D medical imaging data. We evaluate the functionalities and performance of both devices through a user-study in which surgeons and residents assessed the visualization of 3D computer-generated anatomical models. The digital content is obtained through a dedicated medical imaging suite (Verima imaging suite) developed by the Italian start-up company (Witapp s.r.l.). According to our performance analysis in terms of frame rate, there are no significant differences between the two devices. The surgical staff expressed a clear preference for Magic Leap 1, particularly for the better visualization quality and the ease of interaction with the 3D virtual content. Nonetheless, even though the results of the questionnaire were slightly more positive for Magic Leap 1, the spatial understanding of the 3D anatomical model in terms of depth relations and spatial arrangement was positively evaluated for both devices.

## 1. Introduction

Visual augmented reality (AR) encompasses the integration of digital information with the real-world environment. The individual’s perception of the real world is thus enhanced with computer-generated elements that can be ubiquitously added to the natural environment. Unlike virtual reality (VR), which creates a totally artificial scene, AR technology preserves and detects the real view by enriching it with synthetic and locationally coherent elements. 

Historically, Milgram and Kishino proposed in 1994 a continuum model, reported in Figure 1, to describe the flow that goes from virtuality to reality and the various combinations between the two environments in creating extended reality experiences [1]. In the literature, it is common to use the term “mixed reality” (MR) to refer to immersive technologies in which the artificial content interacts with users and the real world [2].

Currently, the broader term XR (extended reality) refers to any immersive experiences that incorporates varying degrees of digital and real information. In particular, the meaning of VR remains the same, while AR is used for static digital information integrated with the real environment and, in the case of a digital–real integration with the possibility of an interaction, the proper term becomes MR (Figure 2) [3].

The first AR headset was created by Ivan Sutherland in the late 1960s [4]; since then, AR/MR visualization has been supported by various output mediums, including smartphones, tablets, computers, and smart glasses. Among them, see-through head-mounted displays (HMDs) seem to be the most efficient means for 3D MR content visualization and performing complex manual tasks under MR guidance. This is because they preserve the user’s egocentric perception of the 3D world, allowing them to interact with it hands-free with no additional attention-shift and mental coordinate-conversion [5].

Optical see-through HMDs (OST HMDs) maintain an almost unaltered direct view of the real world through a special semi-transparent optical combiner onto which the computer-generated content is being projected. The digital content is rendered on a two-dimensional (2D) micro-display positioned outside of the user’s field of view, and collimating lenses, placed between the micro-display and the optical combiner, are used to focus the display content so that it appears at a pre-defined and comfortable distance on a virtual image plane (i.e., the display focal plane). Further than the visualization, simple and instinctive interactions are currently available in high-level MR HMDs; therefore, the interaction model offered by the controller and the associated motion-to-photon latency are significant aspects of an HMD, as a huge latency determines an unnatural and inefficient interaction. Some OST HMDs use input devices such as speech and gesture recognition systems that interpret and translate words and user’s body movements into computer instructions; others employ a dedicated controller with traditional buttons or dedicated movement recognition.

Over the years, MR has been investigated for numerous applications, from architecture to commerce and education. Some examples of the most common application areas comprise gaming, industrial manufacturing, and medicine. 

Although the potential of OSTs for medical-surgical applications is well recognized in the literature, some research studies have identified the main limitations of devices currently on the market. For example, recent studies suggest avoiding the use of commercially available HMDs to perform complex tasks in the peripersonal space (<1 m), such as surgical procedures, owing to their inability to render proper focus cues to stimulate natural eye accommodation responses [6,7].

Despite the technical challenges that still need to be addressed to safely guide the surgical act with sufficient accuracy, naturalness, and comfort for the surgeon, current MR technology is already deemed as a useful asset during diagnostic imaging investigation and medical images visualization [8,9]; this is exactly the purpose of “Verima” (by Witapp s.r.l, Florence, Italy), an application to automatically build and visualize anatomical 3D models starting from medical volumetric dataset as computed tomography (CT). The viewer is designed both for smartphones/tablets (Verima Viewer AR) and binocular HMDs (Verima Viewer MR), while the 3D model reconstruction is based on a deep learning approach [10]. 

The viewer allows the user to manipulate 3D anatomical models and observe the anatomical structures from any perspective and, in particular, in the case of binocular HMDs, thanks to the possibility to induce realistic stereopsis and motion parallax, potentially providing the viewer with a strong sense of depth perception and the immersion of the virtual content in the real environment, with respect to traditional stand-up or hand-held displays. The rationale behind the implementation of this application is to ease the visualization and the 3D immersive exploration of the medical dataset, thus improving the viewer’s spatial understanding of the patient’s anatomical morphology with respect to the same visualization provided on traditional 2D monitors.

In any case, the user perception is hardware dependent and, in this study, we focused on the technological and functional analysis between two of the most popular HMDs, Microsoft HoloLens 2 and Magic Leap 1, to outline the differences and determine which one is more suitable for the visualization of the 3D content associated with the radiological images provided by the Verima suite. 

In this paper, we first analyze the main technological features of the two HMDs, reporting a technological comparison of the performance of the two devices for the target medical application. Then, we report the results of a qualitative evaluation performed by surgeons and residents working at the Cisanello University Hospital (Pisa, Italy) who were recruited to test the application for the visualization of 3D anatomical models on both headsets.

## 2. Materials and Methods

### 2.1. Verima Suite

Verima is a software suite designed for clinicians and medical staff allowing for the generation of realistic anatomical 3D models from diagnostic images (CT and MRI datasets) and for their AR/MR visualization. The AR/MR visualization can be provided via smartphones/tablets (Verima Viewer AR) and HMDs (Verima Viewer MR). The intended use of the suite is to facilitate clinical decision-making as well as support the healthcare personnel in the relationship with the patient. The suite can be used by surgeons to plan interventions and/or share images with colleagues, students, or even patients [10]. 

VR anatomical models can be composed of multiple levels to differentiate between various tissues, and the transparency of the external tissues can be set so that the user can visualize the internal anatomical structures.

The Verima Suite is composed of multiple integrated software, each one playing a specific role: Verima Tool. It is the PC software that converts medical volumetric DICOM files to surface 3D models through an automatic segmentation system based on a deep learning algorithm trained to recognize different anatomical tissues. The tool allows 3D cases to be shared with the HMD connected to the same local network.Verima Desk. It is a web platform, always accessible from any browser, designed for creating, viewing, and sharing clinical cases in 3D starting from DICOM files.Verima Viewer. It is the software (for HMDs, smartphones, and tablets) that allows the user the AR/MR visualization of the 3D models, thus creating a fusion of real and virtual contexts, in which physical and digital objects coexist and interact in real time. The user is able to zoom in, rotate, and disable some details for a complete case analysis (Figure 3).

This study was conducted using the Verima Tool version 5.1.2.0 on a consumer laptop (Dell Vostro 15) and the Verima Viewer version 5.0.6.999 installed on Microsoft HoloLens 2 and Magic Leap 1.

### 2.2. Microsoft HoloLens 2 and Magic Leap One Tecnological Comparative Description

Here, we provide a comparative analysis between two of the most popular high-level OST HMDs: Microsoft HoloLens 2 and Magic Leap 1. These devices are comparable in many ways regarding the operating principle and the basic design and functions. Both headsets use three-stack diffractive waveguides, each of which diffract the RGB waveband, as optical combiners for merging the light coming from the real-world scene with that emitted by the virtual content rendered by the see-through display. For each eye, the virtual content is rendered on a two-dimensional (2D) micro-display placed outside the user’s field of view and near the user’s eyes. Collimation lenses are placed between the micro-display and the semi-transparent waveguides to collimate the virtual 2D image so that it appears at a comfortable viewing distance on a virtual focal plane [11,12]. Such general purpose OST HMDs are mostly designed for parallel viewing; the virtual content is projected at a fixed focal distance (normally between 2 m and infinity), thus they inevitably generate perceptual conflicts, such as vergence–accommodation conflict and focus rivalry [13,14], when used to interact with objects closer to the viewer’s eyes (i.e., within arm’s reach) [7,15]. The main difference between Magic Leap 1 and Microsoft HoloLens 2 is that Magic Leap 1 generates two virtual images projected at two distinct focal planes, one at 0.5 m and the other at 1.5 m, while HoloLens 2 generates only one at 2 m. This feature could potentially mitigate the perceptual conflicts experienced by the user once the active focal plane was adapted as a function of the distance of the real scene [16], but it also comes at the expenses of a reduced see-through capability of the visor as two stacked triplets of waveguides are now needed for each focal plane.

Another hardware difference between the two HMDs is associated with the position of the processing unit: the unit is integrated into HoloLens 2 (Figure 4), whereas in Magic Leap 1, all of the hardware is positioned in a separate case) to attach on the belt or to wear with a crossbody) connected to the glasses via cable (Figure 5).

As said, in HoloLens 2, the processing unit is integrated and there is no additional hardware other than the headset itself: unlike Magic Leap 1, the navigation does not rely on a remote controller, but is totally based on voice and hand gesture commands, making it a completely standalone device. 

A notable limitation of Magic Leap 1 is associated with the display eye relief (i.e., the supported eye-to-waveguide distance), which is not compatible with wearing prescription glasses. Therefore, when using Magic Leap 1, people with vision impairment are forced to wear contact lenses or to buy expansive prescriptions inserts that are available for single vision only and prescriptions with total power in the following range: SPH −7.5 to +3.0 | CYL: −4 to 0 | total power (SPH + CYL): −7.5 to +3.0 [17]. 

Additionally, the frontal part of the display of HoloLens 2 is liftable without having to completely remove the attachment headband. This feature is rather useful to improve the whole usability of the system during prolonged uses and manual tasks. 

The two devices feature six-degree of freedom multi-sensor SLAM capabilities [18]. 

The image quality offered by Magic Leap 1 turns out to be clearer owing to the greater color contrast and lower sensitivity to light. In fact, the Microsoft HoloLens 2 display is strongly susceptible to sunlight and bright lights and, under these conditions, the virtual objects appear less markedly evident. However, and as anticipated, this is mostly because of the Magic Leap optics, which blocks 85% of real-world light (only 15% transmissive), whereas HoloLens 2 has 40% transmissivity.

Both devices are equipped with excellent processors and GPU, but HoloLens 2 is also provided with a customized processing unit that runs all of the computer vision algorithms (head tracking, hand tracking, eye gaze tracking, spatial mapping, and so on) on the device [19].

As shown in Table 1, the field of view (FOV) of the two devices is similar; the FOV of HoloLens 2 is larger than that of its predecessor, but still far from covering the entire human field of vision (Figure 6).

Major technical specifications that may influence device performance for the chosen medical application are summarized in Table 1.

### 2.3. Methodology for Quantitative Performance Comparison

We performed quantitative tests to evaluate the performance of the two devices for the specific medical/surgical application addressed in this paper. 

Regarding the quantitative tests, the framerate in different conditions was evaluated through an ad hoc version of the Verima Viewer. The framerate was shown on the Verima main menu during the visualization of three liver models with different complexity levels in terms of the total number of triangles and the number of segmentation levels (sub-models) included in each file:CASE 1 Liver with kidneys—1,675,664 triangles and 6 levels;CASE 2 Liver 2—250,565 triangles and 57 levels;CASE 3 Liver 3—167,762 triangles and 5 levels.

### 2.4. Methodology for the User Study

We also performed a user-study analysis to qualitatively evaluate the following: the 3D content visualization quality, the interaction modality, the whole HMD ergonomics, and the understanding of the 3D anatomical model in terms of depth relations and spatial arrangement.

#### 2.4.1. Subjects

Twelve volunteers were recruited from medical personnel of the Cisanello Hospital (Pisa, Italy), including surgeons and resident surgeons. All participants claimed to have no prior experience with MR technology. Table 2 reports the demographics of the participants, including people aged between 28 and 55, with normal visual acuity or corrected-to-normal visual acuity with the aid of glasses or contact lenses. 

#### 2.4.2. Protocol of the User Study

The tasks consisted of viewing and interacting with the above-mentioned liver models (example in Figure 7) using the Verima Viewer. Tests were performed on Microsoft HoloLens 2, Magic Leap 1, and on a laptop as a gold standard.

Participants experienced the three devices in a random order to avoid any bias; during the test, the participant could freely zoom in, zoom out, and rotate the 3D model; change the point of view; and select/deselect the levels of segmentation to visualize/hide the different anatomical structures. 

At the end of the experimental session, participants were administered a short questionnaire on their experience and sensations during the test. The questionnaire includes two sections: a first part consisting of a three-point Likert questionnaire and a second part with a multiple-choice question, about the potentiality of the proposed MR visualization, and an open-ended question concerning any problems experienced in the use of each HMD. The Likert questionnaire includes four items assessing the quality of visualization, ease of interaction, comfort, and use of anatomical information. 

#### 2.4.3. Statistical Analysis

The results of the Likert questionnaire are reported in terms of median with dispersion measured by interquartile range (i.e., IQR = 75°–5°). 

The Friedman test was selected to study the qualitative data and then Dunn post-hoc tests were performed to highlight the differences between the three devices. 

The statistical analysis was performed with Matlab (version R2021b) and a *p*-value < 0.05 was considered statistically significant.

## 3. Results

### 3.1. Results of the Performance Comparison

Tests performed with Verima showed that the two MR devices have similar performance in terms of framerate when the app is running without any loaded 3D model. 

The data summarized in Table 3 show that, surprisingly, there is no apparent correlation between the number of triangles and the frame rate. Meanwhile, at least for HoloLens 2, the frame rate decreases as the number of levels increases. Furthermore, with Magic Leap 1, the frame rate difference between the stationary and moving use condition is greater. 

In fact, when a moving user looks at the VR models through HoloLens 2, they seem to be well-anchored in space, whereas with Magic leap, there is a slight, but noticeable, jitter of the virtual objects, which is a perceivable drift of the 3D models from their position.

### 3.2. User-Study Results

The following section reports the results of the user-study. Table 4 summarizes participants’ feedback collected by the administration of the three-point questionnaire about the quality of visualization, interaction, comfort, and overall use of each device.

All participants were able to properly manage the headsets and interact with the 3D models (rotate, zoom, and move), except for one subject who was unable to properly wear Magic Leap 1 with his prescription glasses, thus his answers were excluded from the statistical analysis.

The results collected, in terms of median and interquartile range, immediately show how the surgical staff strongly appreciated Magic Leap 1 HMD in all of the investigated aspects. 

The Friedman test highlights statistically significant differences between the three displays for items 1, 2, and 4. As for items 1 and 4, based on the post-hoc test, the preference expressed toward Magic Leap 1 over the traditional monitor is statistically significant, while there is no statistically significant difference in answering tendencies between the score received by HoloLens 2 and the other two devices. The subjects agreed that Magic Leap 1 offers a better quality of visualization and understanding of the 3D anatomical model over the traditional monitor.

Regarding item 2, the results of the post-hoc test show a statistically significant difference between the scores received by the two HMDs, while they do not highlight statistically significant aspects between the laptop and the other devices. The participants appreciated the remote controller navigation of Magic Leap 1 over the gesture commands of HoloLens 2. Indeed, the navigation modality proposed by Microsoft, although innovative, requires more experience before gaining confidence in the execution.

Regarding item 3, about “the ergonomics of the display according to comfort and/or posture”, there is no statistically significant difference (*p*-value > 0.05) among the three devices, but it can be noticed that the subjects express a mostly positive opinion about this feature for all of them.

In the last section of the questionnaire, we asked the subjects in which contest they would like to use the MR visualization of medical images in the future. The participant could choose from multiple options and the collected answers are reported in Figure 8.

The written comments by the participants agree on the following aspects.

The FOV of both devices was compatible with the visualization of the 3D anatomical models. However, in Magic Leap 1, the lateral peripherical vision of the viewer is obstructed by the frame; this feature can be considered convenient, for an immersive experience, but disadvantageous for those who want to preserve their natural vision of the real world.

As for the wearability of the two devices, Microsoft’s product is heavier on the head, but Magic Leap 1 on the other hand also revealed limitations: the ‘lightpack’ worn around the shoulders limits rapid movements, if it is not fixed in a stable manner, because it starts to oscillate when the wearer moves.

## 4. Discussion

### 4.1. Comparison with the State-of-the-Art

Recently, a systematic review has been presented about the usage of the first-generation HoloLens within the medical domain [20] as it “*emerges as major driving force in medical AR research in the past years*”. In the review, the authors analyzed 217 works spanning a timeframe from the HoloLens release in 2016 until 2021, highlighting promising applications and the most popular research directions. According to the review, the bulk of research focuses on supporting surgeons during interventions. Yet, the consensus is that accuracy and reliability are still too low to replace conventional guidance systems. On the other hand, the second-most common target use is on AR-enhanced medical simulation platforms, for which HoloLens has emerged as a promising technology for its ability to provide an improved perception and understanding of the human anatomy during pre-interventional planning and learning phases. 

In the context of neurosurgical training practice and training, another recent review work has extensively analyzed the latest applications of AR devices [21]. The authors acknowledge that AR could represent a valuable asset in neurosurgery and, more broadly, in image-guided surgery, permitting surgeons to continuously maintain their attention on the surgical field. 

Despite that, both of the reviews share the consensus that accuracy and reliability are still too low to replace conventional surgical guidance systems, whereas their use as surgical simulator or surgical proctoring [22] represents, at the current state of technology, a low-risk scenario where the pros outnumber the cons. This aspect was demonstrated by a plethora of studies that have shown a “*positive association between the use of VR/AR in surgical training and skill acquisition in terms of improving the speed of acquisition of surgical skills, surgeon’s ability to multitask activities, the ability to perform a procedure accurately, hand-eye coordination and bimanual operation*” [23]. 

In conclusion, AR in surgical education is feasible and effective as an adjunct to traditional training with “Microsoft HoloLens showing the most promising results across all parameters and produced improved performance measures in surgical trainees” [24].

### 4.2. Study Limitations and Results Discussion

As already outlined, there is no apparent correlation between the number of triangles and the frame rate. While, at least for HoloLens 2, the frame rate decreases as the number of levels increases. These surprising results could be determined by the Verima suite and not only by the different hardware tested. It is clear that, in general, with bigger and complicated 3D models, the frame rate in both stationary and moving conditions should decrease and it would be useful to define a threshold (currently still unknown) able to guarantee a fruitful visualization of the virtual anatomy. 

Additional user tests should be required to obtain a statistical significance for the answers to item 3 of the user study questionnaire about ergonomics. In any case, this question should be split to acquire a separate opinion at least on the wearability over the head and on the presence of the backpack (in case of Magic Leap 1).

## 5. Conclusions

The surgical staff expressed their preference for Magic Leap 1 because of the better visualization quality and the ease of interaction with the 3D virtual content. Although the HoloLens 2 gestures may be convenient for avoiding the need for a controller, surgeons and residents found Microsoft’s hand gesture to be more complicated in handling virtual objects: familiarizing themselves with HoloLens 2 takes longer than with Magic Leap 1, and some subjects experienced shoulder fatigue from prolonged use of the device. In any case, the understanding of the 3D anatomical model in terms of depth relations and spatial arrangement is positively evaluated for both devices, even though the score is greater for Magic Leap 1.

The evolution of MR technologies has made possible the development of many applications for HMDs. This work investigates the deployment of a medical/surgical application (the Verima suite) on two commercial OST MR headsets: Magic Leap 1 and Microsoft HoloLens 2. The objective is to comparatively evaluate which headset ensures the best features and performances in the visualization and spatial inspection of 3D medical models.

Both devices use the same operating principle and have similar hardware specifications and, according to our performance analysis, in terms of frame rate, there are no significant differences between them.

As emerged from our user study, despite the limitations identified in current technologies, surgeons and residents perceive the potentiality of the proposed MR visualization to be useful for various surgical tasks including preoperative planning and surgical training purposes, which are two of the applications mostly explored in the literature.

Nonetheless, considering the result obtained with this study, we can argue that Verima viewer on Magic Leap 1 offers the user an easier and more efficient MR experience than HoloLens 2.

## Figures and Tables

**Figure 1 sensors-23-03040-f001:**
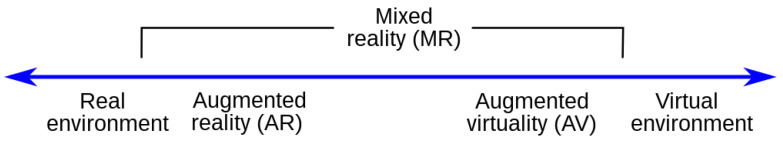
Milgram and Kishino reality–virtuality continuum.

**Figure 2 sensors-23-03040-f002:**
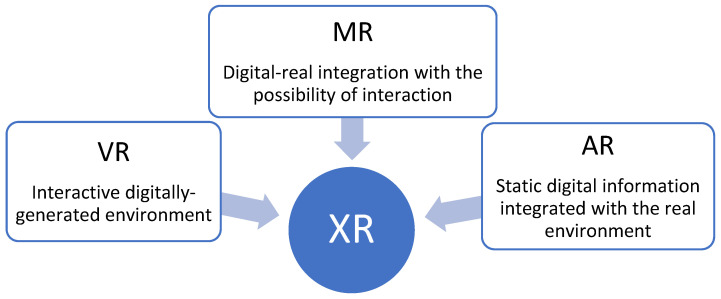
Representation of the collective term extended reality (XR).

**Figure 3 sensors-23-03040-f003:**
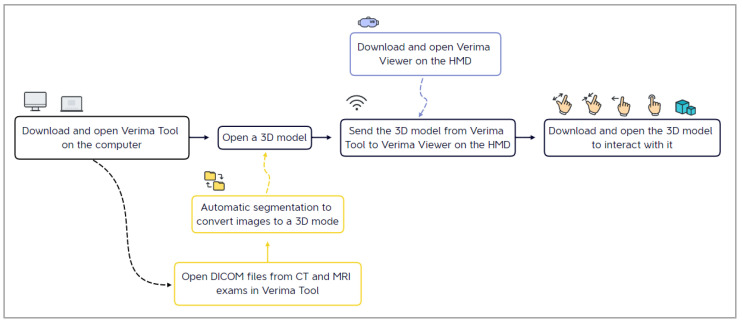
The main steps for the visualization of a surgical case on an HMD, starting from the elaboration of radiological images.

**Figure 4 sensors-23-03040-f004:**
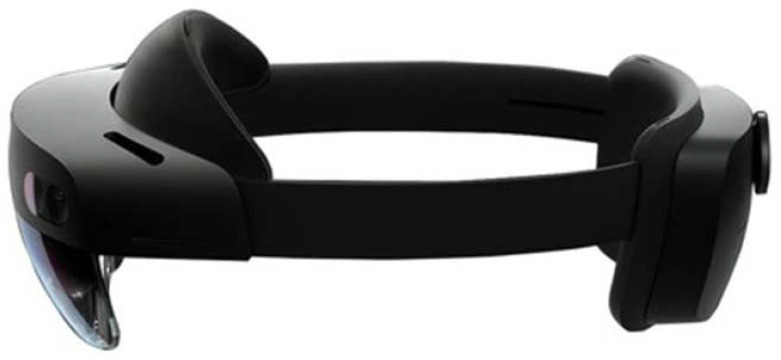
HoloLens 2 headset with the computing unit on the back.

**Figure 5 sensors-23-03040-f005:**
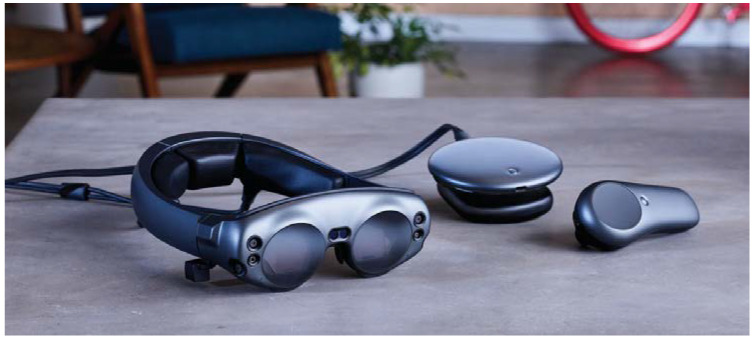
Magic Leap 1 hardware. From left to right: lightwear, lightpack, and control.

**Figure 6 sensors-23-03040-f006:**
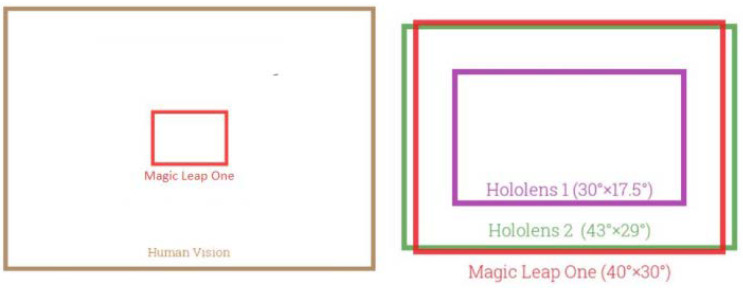
Field of view comparison.

**Figure 7 sensors-23-03040-f007:**
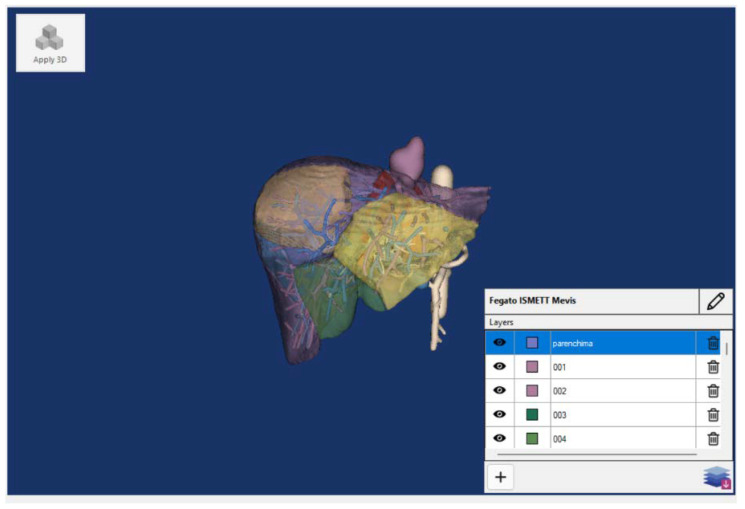
A liver 3D model for the qualitative evaluation and shown on the Verima Tool.

**Figure 8 sensors-23-03040-f008:**
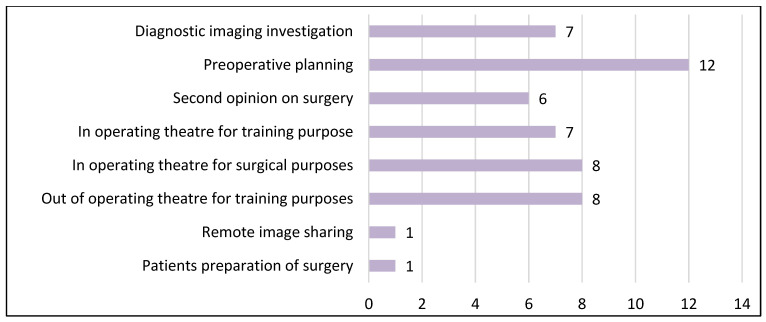
Subject’s answers to future useful uses of MR visualization (multiple choice).

**Table 1 sensors-23-03040-t001:** Overview of major specifications that may influence device performance for the chosen medical application.

	HoloLens 2	Magic Leap 1
Processor	Qualcomm Snapdragon 850	Nvidia Parker SoC
GPU (HPU)	Qualcomm Adreno 630 (Second-generation custom-built holographic processing unit)	Nvidia Pascal 256 CUDA
Operating System	Microsoft OS	Lumin OS
Resolution	2K 3:2 as declared by Microsoft	1280 × 960
Field of View	52° diagonal, 43° horizontal, and 29° vertical	50° diagonal, 40° horizontal, and 30° vertical
Focal Planes	Single fixed at 2 m	Double fixed at 0.5 and 1.5 m
SLAM	6 DoF	6 DoF
RAM	4 GB	8 GB
Storage (ROM)	64 GB	128 GB
Weight	566 g	316 g (without the controller)

**Table 2 sensors-23-03040-t002:** Demographics of participants in the user study.

General Info	Value
Profession/position held (surgeons; residents)	(8; 4)
Medical specialties (general surgery; liver surgery)	(10; 2)
Gender (male; female; non-binary)	(9; 3; 0)
Age (min; max; mean; STD)	(28; 55; 40; 10)
Experience (in years) in surgery (min; max; mean; STD)	(2; 25; 13; 9)

**Table 3 sensors-23-03040-t003:** Frame rate values with the Verima suite.

	HoloLens 2	Magic Leap 1
No running app	60 fps	60 fps
Verima Main Menu	25 fps	25 fps
CASE 1 (1,675,664 triangles and 6 levels)	15 fps (Standing still) 11 fps (Moving)	12 fps (Standing still) 4 fps (Moving)
CASE 2 (250,565 triangles and 57 levels)	10 fps (Standing still) 6 fps (Moving)	25 fps (Standing still) 16 fps (Moving)
CASE 3 (167,762 triangles and 5 levels)	25 fps (Standing still) 15 fps (Moving)	18 fps (Standing still) 8 fps (Moving)

**Table 4 sensors-23-03040-t004:** Questionnaire results about the comparison between different displays (vote from 1 = the lowest to 3 = the highest in terms of agreement). The central tendency of responses is presented by the median, with the dispersion presented by the interquartile range (IQR).

Item	Median (IQR)			*p*-Value
	HoloLens 2	Magic Leap 1	Laptop	
1. The 3D model visualization quality is good	2 (0.5)	3 (0)	1 (1)	0.0061
2. The interaction with the 3D model is easy	1 (1)	3 (0.5)	2 (0)	0.0015
3. I’m satisfied with the display ergonomics, in terms of wearability and posture comfort	2 (1)	3 (1)	3 (1)	0.6065
4. The understanding of the 3D anatomical model in terms of depth relations and spatial arrangement is good	2 (0.5)	3 (0.5)	2 (0.5)	0.0316

## Data Availability

The data presented in this study are available upon request from the corresponding author.

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
