# Peer review of "Magic Leap 1 versus Microsoft HoloLens 2 for the Visualization of 3D Content Obtained from Radiological Images"

_sensors, 2023, doi:10.3390/s23063040_

Round 1

Reviewer 1 Report

Zari et al. present a detailed comparison between Magic Leap 1 and HoloLens 2 for the application of 3D radiological image visualizations to test the mixed reality potentials in medical practice. They use a qualitative study to analyze surgeons' reactions to mixed reality devices to shed light on the application of AR and VR to medical images.

However, I have the following two major concerns:

  1. Since the user study is based on people's reactions and options about devices, they need to take the VR/AR experience level of surgeons into consideration. For example, people may like one device because they have used it before or because they happen to know how to use it. This needs to be included in the analysis.

  2. Besides the qualitative study, I think they can include more quantitative small tasks into the study; actually, they can create the dataset and baseline tasks to measure surgeons' performance. For example, ask surgeons to select an assigned region and count how many seconds they will use in 3 tries to exclude randomness. I hope more comparisons like this can be included.

Besides some small errors they need to update,

  1. Abstract uses 3 paragraphs; try to combine them into one since the last paragraph is just 1 sentence. It is a good writing style to combine them together.

  2. Indentation is wrong in the following lines:

    1. Line 20

    2. Line 68

    3. Line 83

    4. Line 104

    5. Line 133

    6. Line 155

    7. Line 159

    8. Line 222

    9. Line 233

    10. Line 254

    11. Line 272

    12. Line 278

    13. Line 292

Reviewer 2 Report

Please see the attached comments.

Reviewer 3 Report

In my opinion, after reading the paper, it looks interesting. It is sufficient to define the purpose and scope of the study. The following questions should be discussed.

1. It is recommended that the abstract should be compressed into one paragraph. In addition, the contributions of this work should be highlighted to enhance the readability of the article.

2. Different studies on this topic are available before. What is the difference from the others? It should be emphasized.

3. It is not sufficient to explain the results. They need to be disclosed.

4. The references should be updated to illustrate the latest development in the corresponding field.

Reviewer 4 Report

- Add XR as the umbrella of AR and VR, and provide visualization (graph. image, etc.)

- Why this research is conducted? Are there any necessities?

- The methodology section is missing, please add the methodology section to explain why are the authors doing this comparison.

- Which one is more popular in the medical sector, Hololens or Magic Leap?

- Is the popularity align with the results?

- There is no limitation section in this paper

- Add more journals as references

Round 2

Reviewer 2 Report

As responded by authors, they confirm that there is no technological novelty, but offer an user study result, which is not novel enough for publication in this current version.

Author Response

We didn’t write that our user study doesn’t bring novelty.

We didn’t perform a technological development, given that the goal of our scientific paper is to offer interesting and useful objective and subjective data (through a structured user study) about two different technologies produced for the same intended use.

Reviewer 4 Report

Please add a Limitation section that explains the drawbacks of your research. You said they were scattered in the discussion and conclusion sections, however, I encourage you to write them in the special section. It can help readers and future researchers.

Please add more references to reflect the depth and breadth of your understanding of the topic you are researching. 19 references are very few for a Q1 journal.

Author Response

Thanks for your comment.

As suggested, we added 5 new references to improve the scientific value of this paper.

Furthermore, study limitations (and possible improvements) are now in a specific section as required.